# An Exploratory Investigation of UAS Regulations in Europe and the Impact on Effective Use and Economic Potential

Ahmed Alamouri [1,*] , Astrid Lampert [2] and Markus Gerke [1]

1 Institute of Geodesy and Photogrammetry, Technical University of Braunschweig, 38106 Braunschweig, Germany; m.gerke@tu-braunschweig.de

2 Institute of Flight Guidance, Technical University of Braunschweig, 38106 Braunschweig, Germany; astrid.lampert@tu-braunschweig.de

* Correspondence: a.alamouri@tu-braunschweig.de

**Abstract:** Unmanned aircraft systems (UAS) have rapidly become more common in various applications. At the same time, the need for a safe UAS operation is of great importance to minimize and avoid risks that could arise with the deployment of this technology. With these requirements, UAS regulators in the European Union (EU) are making large efforts to enable a reliable legal framework of conditions for UAS operation to keep up with new capabilities of this technology and to minimize the risk of property damage and, most importantly, human injury. A recent outcome of the mentioned efforts is that new EU drone regulations are into force since 1 January 2021. In this paper we aim to provide a sound overview of recent EU drone regulations and the main changes to the rules since the first wave of regulations adopted in 2017. We highlight how such new rules help or hinder the use of UAS technology and its economic potential in scientific and commercial sectors by providing an exploratory investigation of UAS legal frames in Europe. An example of the impact of legislation on the operation of one particular UAS in Germany is provided, which has been in use since 2013 for atmospheric research.

**Keywords:** EU drone regulations; UAS operational categories; UAS classes; risk assessment

## 1. Introduction

In Europe, UAS regulations have varied substantially with large differences of requirements for flight permissions developing with the upcoming technology, and therefore UAS operations are constantly changing with respect to needs and laws of EU member states. Starting with terminologies in regulations, as a simple instance, EU countries use different terminologies in regulations; e.g., unmanned aircraft system (UAS), unmanned aerial vehicle (UAV), drone and remotely piloted aircraft (RPA). For the purpose of uniformity, in this paper we use the term UAS, and include all types of unmanned aircraft systems. However, the term UAS refers to aircraft which are developed to operate without a human operator onboard [1]. Over the previous decade, UASs have developed rapidly and become valuable for various scientific and commercial applications. This can be seen by observing the UAS market, which is growing rapidly, with estimated demand in Europe at 10 billion euro annually until 2035; and might be over 15 billion euro annually by 2050 [2]. In various UAS based application fields such as topographic mapping, infrastructure maintenance, construction surveillance, inspection, etc., these vehicles serve primarily as a platform for sensor payloads which can be an optical camera, a laser device, a synthetic aperture radar, etc. In such applications, a key challenge is how to achieve a safe UAS navigation. To reach the best safety level, it is requested to minimize the risks to other airspace users as well as to both persons and property on the ground. UAS risk avoidance or minimization requests a clear presence of UAS legalization which defines legal frameworks for UAS planning and operation. The legalization should handle with common problems arisen in UAS regulations such as time of operation, e.g., within/outside the rush-hour [3], safety

and administrative issues that hinder the desired flexibility in the execution of administrative processes and impede the widespread utilization of the UAS technology [4]. From this point of view, some national and international authorities and organizations for EU aviation started to update and modernize the first wave of regulations adopted in 2017 with focus on: (a) keeping up with recent technological developments and new capabilities of UAS, (b) seeking to accommodate user demands and (c) increasing safety level during the operation. In this context, the recent outcome of the modernization is that the new EU drone regulations are into force since 1 January 2021, which are seen as a positive step towards UAS rules harmonization in Europe. Nevertheless, applying the new drone rules into national legislation may probably not take place before the end of 2021. This is due to many still open questions related to administrative and technical details that need to be discussed, such as defining responsibilities of local authorities, etc. Of course, this will only be possible with large efforts that can help in converting the European regulations into federal and state acts.

However, this article is motivated by the desire to help understand recent UAS regulations and to assist users in navigating through the administrative and bureaucratic processes to implement an effective legal use of UAS. The aim of this paper is, therefore, to provide a sound overview of recent regulation issues associated with UAS planning and operation. The paper proceeds with background information about the importance of UAS regulations as an indispensable prerequisite that dictates when, where and under which conditions UAS can be operated. Next, the status of the regulations will be discussed. Here, the most important amendments and changes in EU drone regulations since 2017 will be introduced. Finally, we address the impact of new regulations on UAS use and its economic potential by providing an exploratory investigation of UAS legal frames. This includes a global overview and thorough discussion of main regulations and criteria that have to be considered in commercial and scientific sectors, civilian applications such as transport, energy and utilities, agriculture, etc.

As an illustrative example, the procedure required to acquire the flight permission for a particular system, which has been operated since 2013, is provided. The UAS "ALADINA" [4], with a weight of 24.9 kg and a wingspan of 3.6 m, operated by the Institute of Flight Guidance of TU Braunschweig, is applied for atmospheric research, thus for altitudes up to 1000 m. The UAS has been operated on during different years in Germany [5–7], in Benin [8,9] and in Svalbard [10,11].

## 2. Related Work

UAS have become relevant for various applications for many years, e.g., for atmospheric research, UAS applications date back to the 1970s [12]. With the technical progress and the disruptive miniaturization of sensors, also motivated by the telecommunication industry, the capabilities of off-the-shelf-systems, including airframe, autopilot and ground control station, have increased significantly. Such an inexpensive technology has become more valuable in practical uses and opened new opportunities in scientific and commercial sectors. As a result, the UAS traffic in the sky is continuously increasing; and this prompted a focus on the need to increase safety factors for UAS operations. An increasing safety level during UAS operation requests decreasing/minimizing the expected operation risks and hazards that cannot only influence airspace users, but also persons and properties on the ground. At the same time, avoiding/minimizing UAS risks needs a clear legalization framework for UAS planning and operation. UAS legalization has already been discussed in various publications. Literature reviews revealed that authors of relevant publications dealt with UAS regulations from the perspective of one context (e.g., cost, privacy), operational aspects (e.g., operating time, risk avoidance), etc. Furthermore, one can observe that this topic has been mostly addressed at national levels or covered a few states [4].

In terms of national legalization, as an example we refer to the contribution achieved by Cramer and Wieland (2019), who reported about the UAS regulations in Germany and the important changes from 2017 to 2018 in the context of UAS operation and safety [13].

Another contribution, also focusing on the German UAS regulations, has been published by Borst et al. (2020). The contribution discussed UAS operations in German airspace according to the EU rules. It pointed out that Germany has created legal regimes to support defining and realizing regulations for manufacturers and operators of UAS [14].

Prior to 2017, for flight operations of ALADINA, the 16 federal states of Germany were responsible for flight permissions. The application for flight permission was submitted to the respective federal aviation authority where the flights were planned. The following documents were required: documentation of planned flight trajectories and flight times with safe distance to infrastructure and persons, agreement of the owner or tenant of the land where flights should take place, agreement of the natural protection agency, agreement of the regulatory agency ("Ordnungsamt") or the local police, technical description of the UAS including safety aspects and procedures, description of the scientific purpose of the measurements, description of the pilots' experience and insurance certificate. Based on this documentation, the federal aviation authority asked for a statement of the German air traffic control ("Deutsche Flugsicherung"), who finally issued a NOTAM (notice to airmen) warning pilots about the UAS activities in this area and up to the required altitude. Based on this NOTAM, the certificate of the flight permission was granted with certain obligations: besides the safety pilot, a second crew member was required to observe the sky for potential other air traffic. The crew reported the beginning and end of activities to the air traffic control. A contact phone number of the UAS crew was provided, which had to be on stand-by during operation in case of any unforeseeable events. Sometimes restrictions for operation times were imposed.

After 2017, universities were treated as public authorities. No official permission from the aviation authorities was required for the operation of ALADINA and other UAS of the same size and weight class. Flight plans were discussed directly with air traffic control to issue a NOTAM, and, if necessary, a permission of the respective federal environmental agency was required. Since 2021, the federal aviation authorities are again responsible for ALADINA flight permissions. The required documents include a risk analysis according to SORA (see Section 4.3). The exception is if the flights are directly authorized by a legal entity, which can be the Federal Environment Agency, for example, or flights in military restricted airspace.

However, in order for UAS to become an efficient and safe technology in a wide frame, e.g., for public and land management authorities and institutions in Europe, and a reliable base for company investments, a legal use of this technology on EU level is indispensable. Thus, based on a pre-defined and clear legal use, a solid background of the rules can be provided to UAS users to ensure the safety of UAS operation and the efficient use of EU airspace. In this context, we notice that UAS regulators in Europe are increasingly relying on the installation of UAS legal frameworks and airspace classification. An example for a legal framework installation is UAV DACH, which is an association for UAS activities in Germany, Austria, Switzerland and The Netherlands (www.uavdach.org) (accessed on 11 April 2021). It takes an important role in developing and realizing UAS rules for UAS safe operation and navigation. It promotes unmanned aviation in order to increase its acceptance and cares for issues related to safety. Its aim is to satisfy the regulatory relaxation needed by the commercial UAS sector in a way that allows for further drone investment on the condition that the regulatory environment becomes clearer and friendlier towards UAS operation and navigation. In general, airspace classification is similar for EU countries. For instance, the German airspace is divided in controlled and uncontrolled airspace. The controlled airspace is structured in the following categories: C (Charlie), D (Delta), E (Echo) and Controlled zones CTR. Uncontrolled airspace G (Golf) is only usable for UAS without an air traffic control clearance [15]. Uncontrolled airspace usually has an upper boundary of 2500 ft (762 m) or 1000 ft (304.8 m) above ground level, depending on location, and is not available at all around airport control zones.

The demand for uniform legalization is still a cumbersome task for regulators, because regulations must encompass UAS technological developments and new capabilities at the

same time as they occur [16] and, therefore, this leads to continuous updates of regulations over time. However, for countries with existing UAS legalization, rules are constantly being re-evaluated and harmonized; almost all listed regulations have been written or amended within the past years [17]. Currently, many efforts towards harmonizing UAS rules and adopting uniform regulatory standards are being undertaken by the European Commission with a focus on introducing a proposal to integrate all UAS, regardless of their size, into the EU aviation safety framework [18]. A very important step towards EU-wide harmonization is the new EU drone regulations (which came into force on 1 January 2021), which define rules for all EU countries, such as identification of airspace classes, UAS operating categories, etc. In addition, the new EU regulations provide country-specific requirements as national legal standards of the individual EU member states that must also be met.

To this end, it can be seen that the further development and use of UAS technology in Europe greatly depends on regulations governing the use of UAS in different EU-states, and, therefore, we are motivated to highlight the development and impact of UAS regulations according to the policies of EU authorizations involved in developing UAS legal frameworks.

## 3. Development of UAS Regulations

### 3.1. Global Scale Regulations

When dealing with UAS regulations at global scale, it is important to address a brief background of the International Civil Aviation Organization ICAO (www.icao.int) (accessed on 16 April 2021). According to the Chicago convention in 1944, ICAO was founded and is now directed by 193 national governments towards support and manage transport issues. During the second informal meeting concerning the UAS (January 2007) of ICAO, it was concluded that ICAO should serve as a central point for global interoperability and harmonization in terms of the following: "(1) developing a regulatory concept, (2) coordinating the development of UAS Standards and Recommended Practices (SARPs), (3) contribute to the development of technical specifications by other bodies, and (4) identify communication requirements for UAS activity" [1]. However, UAS activities steadily increase and, therefore, UAS operators are a fast-growing group of airspace users which need larger portions of airspace for their operations. From this point of view, in the 40th assembly of ICAO, September 2019, the safe and efficient integration of UAS into global airspace was discussed. As an outcome, it emphasized the importance of reviewing and improving the operational framework of UAS in technical, economic and legal fields [19].

### 3.2. EU Drone Regulations

We highlight in this section the status of the regulations with regard to policies of organizations and authorizations entrusted with developing regulation concepts in Europe. For a better understanding of the UAS regulations status in EU, we briefly review the status of regulations and important issues adopted, before the new EU regulations were approved on 1 January 2021 (Figure 1).

Starting from March 2017, the European Aviation Safety Agency EASA (www.easa.europa.eu) (accessed on 06 April 2021) laid down the first wide wave of regulations focusing on air traffic management and navigation services. This regulation wave classified UAS in different categories according to the maximum take-off mass MTOM allowed for UAS. Hence, it was distinguished between systems with MTOM < 5 kg, between 5–25 kg and over 25 kg, as it is assumed that the risk is significantly related to the potential and kinetic energy of UAS. In the same year, a new concept for basic regulations has been proposed and discussed between the European Council, European Commission, and the European Parliament, to regulate all UAS regardless of their MTOM [20]. In the following year, the focus was on establishing a legal framework for a high uniform level of civil aviation safety in the EU. The aim of the mentioned framework was to improve the overall performance of civil aviation by adopting effective aviation policies in Europe [21]. As a result, rules for

UAS operation and requirements for technology and personnel, including the involvement of remote pilots, have been adopted at the EU level in 2019 [22].

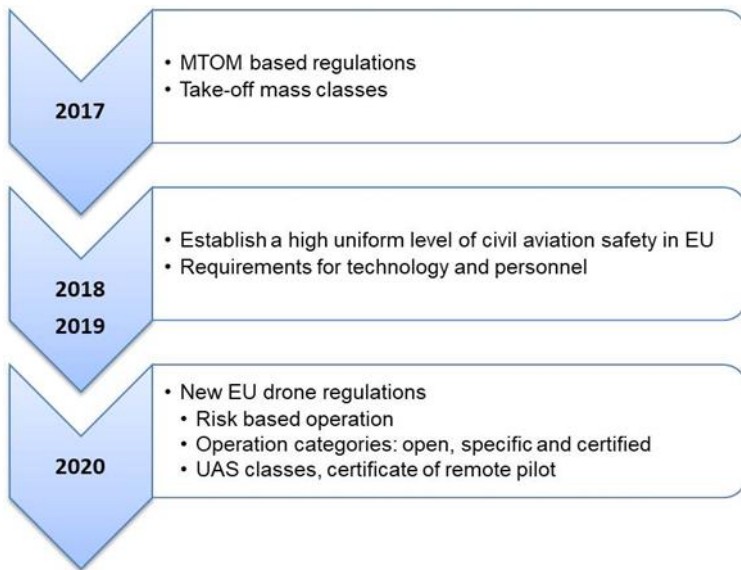

**Figure 1.** Important updates on EU drone regulations from 2017 to 2020.

In a further step, EASA laid down the new UAS rules in December 2020, which are seen as a development step based on the notice of the proposed amendment A-NPA in 2015 [23]. It introduces a thorough overview about two main parts: implementing regulations IRs, (operations of UAS—Regulation EU 2019/947) and delegated rules DRs, (technical requirements for the design and manufacture of UAS—Regulation EU 2019/945). Implementing rules—which are the main focus of this paper—lay down thorough provisions for UAS operations. They provide regulations to personnel and organizations involved in those operations. In this context, and for a safe UAS operation, IRs propose three categories of UAS operations based on the risk the operation is posing to third parties [22]; these are: open, specific and certified (Figure 2).

- UAS operation category "open".

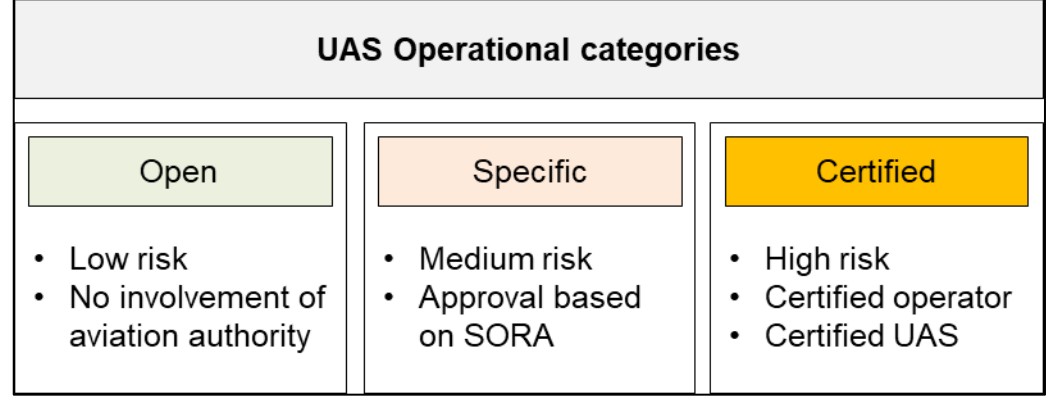

**Figure 2.** UAS operational categories.

The open category is denoted as being low risk during UAS activities. The main advantage, therefore, is the possibility to fly without an operating license that is requested by other categories and needs a certain amount of effort to obtain. According to Article 4—IRs EU 2019/947, "UAS operations are classified in the open category only where the following requirements are met [22]:

- UAS has a class that is set out in DRs EU 2019/945
- UAS maximum take-off mass should be less than 25 kg
- UAS operation is conducted in the visual line of sight VLOS and the UAS is kept at a safe distance of at least 1.5 km from inhabited areas, airports and sensitive zones, and at least 100 m from infrastructure like highways, hospitals, power plants, etc.
- During an operation, UAS do not carry dangerous goods and do not drop any material
- Flying height is limited to 120 m above the surface of the Earth".

From an operational point of view, the new rules divide the open category into three subcategories: A1, A2 and A3, depending on the operation distance to persons, and which result in different restrictions concerning the weight of the UAS and the certification of the pilot. The classification is based on UAS class-identification showing technical properties of UAS and operational requirements to be considered during operational processes. According to this class-identification, UAS are structured into the seven classes C0 to C6 (Table 1).

**Table 1.** UAS classes (C0–C6) according to technical properties and operational requirements. Source: Delegated Rules EU 2020/1058.

| Subcategory | Class | MTOM Incl. Payload | Velocity | Max. AGL | Proof of Knowledge |
|---|---|---|---|---|---|
| A1<br>Fly over people | C0 | <250 g | max 19 m/s | 120 m | Familiar with operation instructions |
| | C1 | <900 g | max 19 m/s | 120 m | Familiar with operation instructions<br>Online training and test |
| A2<br>Fly close to people | C2 | <4 kg | - | 120 m | Familiar with operation instructions<br>Online training and test<br>Certificate "proof of knowledge" (according to German rules) |
| A3<br>Fly far from people | C3 | <25 kg<br><Diameter 3m | - | 120 m | Familiar with operation instructions<br>Online training and test |
| | C4 | <25 kg | - | - | |
| - | C5 | No max. MTOM defined | - | - | |
| - | C6 | | max 50 m/s | - | |

In the subcategory A1, UAS operators—if to fly with class C0—do not need to obtain an operating permit or submit an operating declaration before commencing operations. In addition, UAS operations in A1 shall not be conducted over open-air assemblies of persons. In contrast, remote operation—flying with class C1—requires that remote pilots are familiar with the UAS user manual guide and have passed an online training course. The main difference between C0 and C1 is the MTOM, which is a maximum of 250 g vs. 900 g.

Regarding the subcategory A2, remote pilots must be familiar with the UAS user guide and have to hold a remote pilot certificate. In addition to that, UAS operation must not be conducted over and always at a safe distance from uninvolved persons. In this context, it is essential to have a high safety level, to respect the privacy and environmental requirements.

Finally, by flights in the subcategory A3—fly far from people—remote pilots have to be familiar with the UAS user guide, too. However, the UAS operations must occur in an area where the remote pilot can reasonably expect that no uninvolved persons will be endangered within the range where the UAS will be operated. Within this scenario, UAS pilots have to maintain a horizontal safety distance from public utilities and residential, industrial, or recreational areas.

The technical properties and operational requirements did not define a maximum Above Ground Level (AGL) in the classes C4, C5 und C6, but mentioned that "during

flight, provide the remote pilot with clear and concise information on the height of the UAS above the surface or take-off point" [22].

However, when at least one of the above listed requirements is not met, then the operation no longer belongs to the open category. For instance, when a UAS is operated beyond visual of line. Another example is operating a 15 kg UAS close to a gathering of people; in this case, within A2, this is limited to a maximum MTOM of 4 kg.

- "Specific" category of UAS operations

If, for certain reasons, one or more of the regulations of the open category cannot be complied with, the specific category takes effect because a higher risk can be expected. From this point of view, the specific category laid down rules covering UAS operations presenting a higher risk during flying for which a thorough assessment should be carried out to indicate which measures are needed to keep the operation as safe as possible. In such scenarios, either an operating permit is required or a prior declaration must be made. To obtain an operating license, a risk assessment is necessary and has to be reviewed by the competent authority. This is much more extensive than a simple declaration. According to IRs EU 2019/947—Article 11 [22], an operational risk assessment shall include, but is not limited to "(a) description of UAS operation, (b) proposal for maintaining operational safety, (c) identification of ground and air risks to, for example, uninvolved persons, objects, etc., (d) measures for risk mitigation, (e) technical characteristics of the UAS and (f) competencies of the personnel".

In order to conduct the operational risk assessment required by the abovementioned Article 11 of the UAS Regulation, the Specific Operation Risk Assessment (SORA) can be applied. SORA is the methodology developed by the Joint Authorities for Rulemaking on Unmanned Systems JARUS (http://jarus-rpas.org/) (accessed on 4 May 2021) to perform risk assessment and, therefore, to safely conduct UAS operations [24]. More details about this approach and its concept are available at EASA, 2019/947. The SORA concept and an example for ALADINA will be discussed in Section 4.3.

- "Certified" category of UAS operations.

According to IRs EU 2019/947—Article 6 [22], the operating of UAS missions is considered in the certified operation category if the UAS is certified pursuant to Article 40 of DRs EU 2019/945. In addition, operations can be classified in the certified category, if the competent authority may assess the operational risk such that the operation falls into the certified category. Certifying UAS covers the design, production and maintenance of UAS. It is definitely required if the UAS meets any of the following conditions: "(a) the dimension of the UAS is at least 3 m and designed to be operated over assemblies of people, (b) it is designed for the transport of people, and (c) it is designed to transport dangerous goods and requires a high level of robustness to mitigate risks to third parties in the event of an accident". The certified category includes, in addition to the certification of the UAS itself, the certification of the entire operation, i.e., the operating company, the remote pilots, the maintenance of the UAS, the monitoring of the maintenance Continuing Airworthiness Management Organization (CAMO).

## 4. Impact of New Regulations on UAS Operation

Although the new regulations are seen as a positive step towards harmonizing UAS rules in Europe, they will inevitably lead to new challenges and restrictions that influence UAS operations and uses. In addition, it is expected that the challenges and limitations will not only have an impact on UAS pilots, but also on the manufacturers who are looking for a clear legal framework that helps ensuring a high level of rule compliance. However, the following subsections address important challenges and limitations that might be at least faced in the early stages of applying the new EU drone rules.

*4.1. Challenges with UAS Registration*

New UAS regulations in Europe are considered as an important step to move towards rule harmonization and better accommodating of UAS operations, but it might be feared that UAS owners, pilots, operators and manufacturers will find the regulations cumbersome due to the administrative and bureaucratic complexities in rule interpretation. In addition, the regulations are under development and might change continuously, and this will lead, on the one hand, to instability of UAS legal operational framework and, on the other hand, to some confusion during the achievement of administrative and bureaucratic processes. To clarify the idea, we refer to the registration of UAS operators and certified UAS. According to the Article 14 from EU drone regulations [22], the rules impose that a registration system of UAS and users should be established by EU member states for "UAS whose design is subject to certification and for UAS operators whose operation may present a risk to safety, security, privacy, and protection of personal data or environment". From this statement, it is to understand that the registration process depends on UAS design and risks which are the main factors respected with the defining UAS operation categories (Section 3.2). This means that the intended registration system is dependent on—among other things—the operation category where the UAS should be operated. So, once again, according to Article 14, a registration is required when flying in the open category if the UAS meets any of the following conditions: (a) MTOM is 250 g or more, (b) UAS is integrated with a payload like for instance a sensor that could be used for personal data collection. In contrast, when it is to fly in the specific category, the registration of UAS is mandatory. To this end, the rules have clearly addressed who should register, and when, but realizing the aforementioned registration system in a practical way is still a key challenge. This is due to the fact that each EU member state has to create an online platform for the registration process, which does not yet exist in many EU countries. As a result, the interoperability, mutual access and exchange of registration information might be affected.

Another issue to be respected is the protection of personal data. Within the registration process, personal information should be provided, such as the full name, date of birth, addresses of UAS operators, etc. In many EU countries, data protection laws are strongly enforced and, therefore, data interoperability and mutual access at the EU level are currently unrealistic or at least require further efforts to develop reasonable systems for registration that enable data exchange with a high security level. Of course, this requires databases that document, manage and analyze the collected data. Here, involved specialists, operators and managers have to work closely together to define which parameters and issues are necessary and should be respected in the database design.

*4.2. Visibility and Range Restrictions*

The main criteria of most UAS regulations revolve around the limitations of UAS operations and refer to restrictions of flight missions [4]. To express the idea here, we refer to the visibility and range restrictions in terms of height levels and horizontal distances allowed during UAS operations; namely operating UAS with visual/beyond line-of-sight (VLOS/BVLOS) conditions (Figure 3). According to the recent EU rules, UAS operations classified in the open category are only allowed with VLOS of the remote pilot (Section 3.2). Within this scenario, pilots must maintain continuous visual contact with UAS. VLOS is particularly interpreted to mean up to 500 m horizontally and 120 m vertically [25], but for large and well-visible systems (striking painting, position lights), altitudes up to 1000 or even 1500 m, and a radius of 1.5 km around the operator, have been accepted as VLOS (see measurements in [5,6,11]). In practical usage, UAS are sometimes operated beyond the aforementioned distance limitations as extended visual line of sight (EVLOS). Within EVLOS-based operations, pilots need additional observers or remote pilots to keep continuous visual contact with UAS. Under these conditions, and regarding the economic viability of UAS-based applications, both VLOS and EVLOS scenarios are not the best option for the commercial UAS sector, where flight missions beyond visual line of sight (BVLOS) are hugely valuable due to the fact that they enable UAS to cover long distances

and large areas beyond the visual range of the remote pilot; especially when obstacles such as buildings and mountains could be encountered during VLOS flight missions.

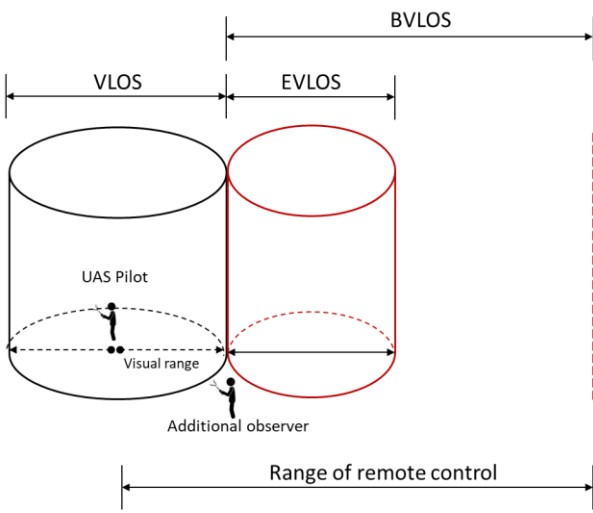

**Figure 3.** Illustration of VLOS and BVLOS—Source [4].

A better understanding of BVLOS potential can be addressed throughout a comparison between typical applications for VLOS and BVLOS (Figure 4). One can observe, in UAS markets, that manufacturers/users are adopting the BVLOS technique for better UAS functionalities and opening new commercial opportunities, for example, first responders, package delivery (already tested by Amazon), inspections, atmospheric science, etc.

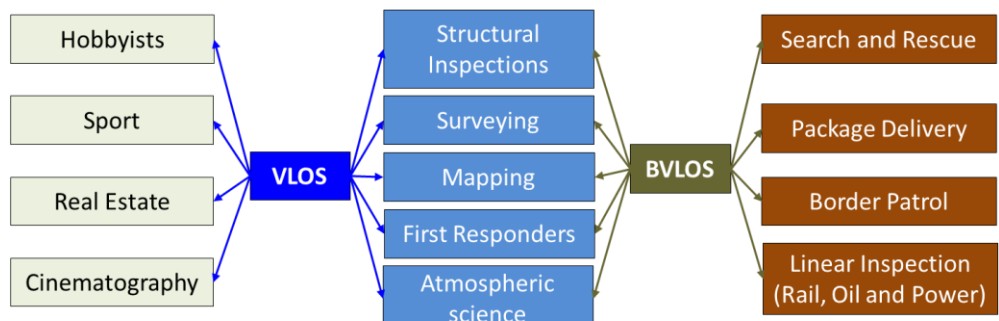

**Figure 4.** VLOS- and BVLOS-based typical applications—modified from [25].

Getting rid of VLOS confinements is possible by operating UAS in the specific or certified categories. It is agreed that operations in these categories can be a solution to achieve BVLOS flights, but at the same time and from a practical point of view, it imposes additional processes and requirements, such as conducting an operational risk assessment, for instance that based on JARUS SORA, which in itself is a complex issue directly affecting UAS operation. This will be highlighted in the next section.

*4.3. Operational Risk Assessment*

The risk assessment of UAS operations reflects the safety level associated with UAS during operating time. A safety level reflection needs to make safety risks measurable and, therefore, risks can be evaluated and controlled. The new EU drone rules adopted a risk assessment as a fundamental issue that is requested for flights in the specific and certified categories. In practice, the assessment of a UAS operation risk is an important step towards creating safe flight missions, but at the same time realizing a reasonable and acceptable risk assessment is still challenging, because it involves different responsibilities

and processes that extend beyond a single agency or organization. Furthermore, it needs to be balanced between various complex factors and data coming from social, technical, political and economic aspects [26]. Collecting and characterizing risks that may occur during UAS operations need to describe, on the one hand, the UAS environment where the UAS is flown, and on the other hand, the risk nature such as the ground risk (e.g., damage of third parties on the ground) or the air risk (e.g., flying into forbidden zones). It dictates having sufficient information, data and resources which are not always available and/or outdated. Currently, the new rules mentioned that the operational risk assessment can be conducted based on the SORA developed by the joint authorities for rulemaking on unmanned systems [22]. The paradigm implemented in SORA is to minimize the impact of a possible drone risk, which means damage to third parties on the ground or in the air; however, this classification is only done based on general rules and estimated from the properties of the UAS type used for the mission, the type of air space, and the population of the area where it is operated.

Key elements of the SORA classification are the "risk" and "robustness". "Risk" is defined according to SAE ARP 4754A/EUROCAE ED-79A as "the combination of the frequency (probability) of an occurrence and its associated level of severity". The term risk is only applied for detrimental events. "Robustness" defines the requirements for UAS operation for different risk classes. The level of robustness can be low, medium or high. Operations with higher risks require higher levels of robustness [22].

For a SORA classification, the concept of operation (ConOps) has to be defined first, comprising technical details, missions, checklists and safety aspects. Technical details include information on the UAS fuselage (dimensions, MTOM, loads, subsystems such as hydraulic systems, brakes, parachute, sensors), performance characteristics (maximum flight altitude, climb and descent rate, air speed, maximum air speed, limitations induced by icing, precipitation, turbulence and other parameters), propulsion system (engine type and number of engines, engine power, electric system, maximum current), control system (flaps, pitch elevator, aileron, actuators), sensors for operation, payload (power supply, impact on flight parameters), navigation, autopilot, flight control, ground control station, detect and avoid system, compliance with geo-fencing, take-off and landing equipment, and implemented functions such as flight termination system or automatic recovery system.

The risk is subdivided into the "ground risk" and the "air risk". The ground risk class (GRC) defines the risk to uninvolved persons on the ground in the case of control loss of the UAS. According to SORA, there are five different scenarios determining the ground risk:

- VLOS or BVLOS in controlled areas (such as military areas);
- VLOS in sparsely populated areas;
- BVLOS in sparsely populated areas;
- VLOS in populated areas;
- BVLOS in populated areas.

Therefore, the operation site and area, called volume of operations, have to be well defined in advance. Further, air speed, mass and wingspan are important. The ground risk class can be determined based on the technical specifications (Table 2).

The typical air speed and mass are used to calculate the expected typical kinetic energy. The wingspan is used as the "maximum UAS characteristics dimension". As an example, the risk of operating ALADINA is assessed in the following according to SORA. For the estimation of the typical kinetic energy and the resulting ground risk class (GRC) the following values are assumed:

- Mass m = 25 kg; air speed VTAS = 25 m/s; wingspan b = 3.6 m;
- The kinetic energy is calculated to be 7.8 kJ;
- With a wingspan of 3.6 m, this results in risk class <8 m, <1084 kJ;

As a next step, the flight scenario needs to be chosen. ALADINA is operated up to altitudes of 1000 or 1500 m, with a radius of 1500 m around the operator. The measurement site is usually chosen in a way that no villages or infrastructure are overflown. In the past,

altitudes have been accepted as VLOS, as ALADINA is a large aircraft with striking colors and illumination. Therefore, the scenario is "VLOS in sparsely populated environment", resulting in a preliminary ground risk class of 4 according to Table 2.

**Table 2.** Determination of ground risk class according to SORA. This table is from JARUS guidelines on Specific Operations Risk Assessment (SORA), 2019, p. 20.

| Intrinsic UAS Ground Risk Class | | | | |
|---|---|---|---|---|
| Max UAS characteristics dimension | 1 m | 3 m | 8 m | >8 m |
| Typical kinetic energy expected | <700 J | <34 kJ | <1084 kJ | >1084 kJ |
| **Operational scenarios** | | | | |
| VLOS/BVLOS over controlled ground area | 1 | 2 | 3 | 4 |
| VLOS in sparsely populated environment | 2 | 3 | 4 | 5 |
| BVLOS in sparsely populated environment | 3 | 4 | 5 | 6 |
| VLOS in populated environment | 4 | 5 | 6 | 8 |
| BVLOS in populated environment | 5 | 6 | 8 | 10 |
| VLOS over gathering of people | 7 | | | |
| BVLOS over gathering of people | 8 | | | |

There are different options for ground risk mitigations according to SORA (Table 3), and each sequence reduces the ground risk class, depending on the level of robustness.

**Table 3.** Ground risk mitigation for different robustness levels—JARUS guidelines on SORA, p. 21.

| Mitigation Sequence | Mitigations for Ground Risk | Robustness | | |
|---|---|---|---|---|
| | | Low/None | Medium | High |
| M1 | Strategic mitigations for ground risk | 0: None −1: Low | −2 | −4 |
| M2 | Effects of ground impact are reduced | 0 | −1 | −2 |
| M3 | An Emergency Response Plan (ERP) is in place, operator validated and effective | 1 | 0 | −1 |

The strategic mitigation includes measures to reduce the risk for persons on ground, e.g., by establishing a horizontal buffer zone of the same dimension as the flight altitude that cannot be entered by third parties. The effects of ground impact can be reduced, e.g., by applying an emergency parachute. Depending on the robustness of the system (e.g., redundancy), this can significantly reduce the ground risk. The third possibility to reduce the ground risk is an emergency plan for the case of loss of control. This emergency plan needs to be well established, as a missing plan can even enhance the ground risk. Strategic mitigation—in this case, a 1:1 buffer zone of at least 1500 m to the next village, and a valid emergency response plan—reduce the ground risk class of ALADINA to 3.

The air risk class (ARC) is an index for the risk of a collision with a manned aircraft. Four classes of air risk are defined: for ARC-a, the risk of collision is so low that no strategic measures have to be taken. The risk increases up to ARC-d. The air risk class depends on the air space and expected air traffic and can be determined with the flow chart of Figure 5. For ARC-b to ARC-d, the risk can be mitigated by several methods.

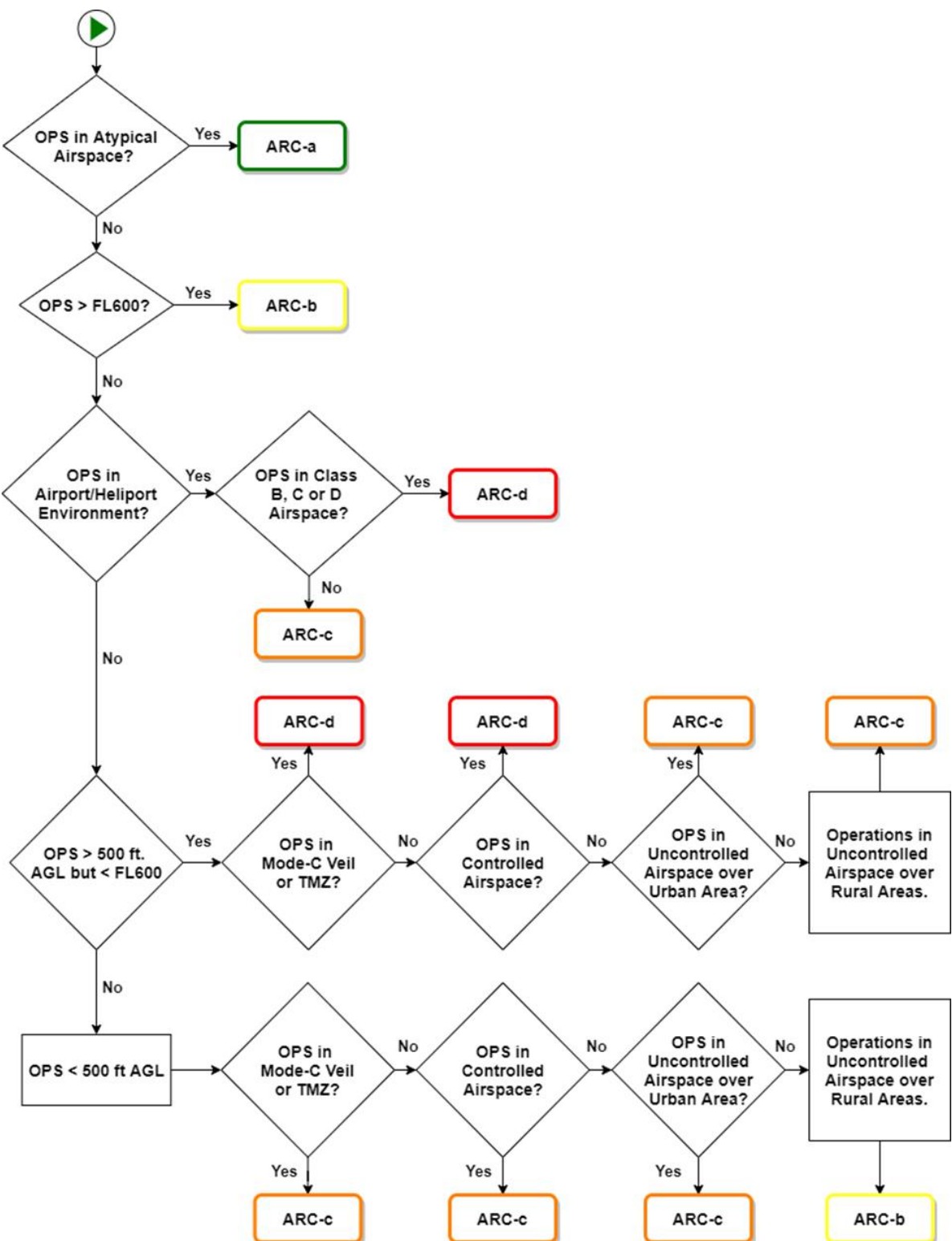

**Figure 5.** Criteria for air risk class. Source: JARUS guidelines on SORA, 2019, p. 23.

As ALADINA is in an atypically controlled airspace during flight (air space reserved by NOTAM), it falls into the lowest air risk category ARC-a. Additionally ALADINA can be equipped with an ADS-B receiver, which receives position signals from other aircrafts and further reduces the risk of collision.

Based on the values of air risk and ground risk, the "Specific Assurance and Integrity Levels (SAIL)" are determined (Table 4). The SAIL parameter determines the requirements for the Operational Safety Objectives (OSO).

**Table 4.** Specific Assurance and Integrity Levels (SAIL) according to SORA (JARUS, p. 27). For the example of ALADINA, which is discussed in the text, the value is marked in yellow.

| SAIL Determination | | | | |
|---|---|---|---|---|
| | Residual ARC | | | |
| **Final GRC** | **a** | **b** | **c** | **d** |
| ≤2 | I | II | IV | VI |
| 3 | II | II | IV | VI |
| 4 | III | III | IV | VI |
| 5 | IV | IV | IV | VI |
| 6 | V | V | V | VI |
| 7 | VI | VI | VI | VI |
| >7 | Category C operation | | | |

The assessments described above for ALADINA result in a SAIL Level of II. This level requires the implementation of an "optional" or "low minimum security" standard. The difference to the requirement "Medium" is the quality of the operational procedures for normal operation, deteriorated systems, human errors, and adverse operating conditions.

Requirements for robustness: This section is based on JARUS, Annex E—Integrity and assurance levels for the Operation Safety Objectives (OSO), JARUS guidelines on Specific Operations Risk Assessment (SORA), 2019. There are different aspects that have to be analyzed for determining the correct level of integrity and assurance. The full specifications are provided in the JARUS guidelines on Specific Operations Risk Assessment (SORA), 2019. A few examples of the aspects that have to be considered are the following:

- Competent or approved operator;
- Manufacturing of the UAS by competent entity;
- Competent maintenance;
- Compliance with design standard;
- System safety and reliability;
- Suitable command, control, and communication practices;
- Inspection of the UAS for compliance with the ConOps;
- Definition, validation and compliance with technical requirements and concerning human failure;
- Training and ability of the crew to deal with abnormal situations (covering technical and human failure);
- Safe recovery from technical problems;
- Procedures to deal with external disturbance;
- Appropriate external support for the mission;
- Coordination between the crew members;
- Operational crew (fit in physical and psychological sense);
- Automatic protection of the UAS against human failure;
- Handling of human failure;
- Human–machine interface;
- Detection and avoidance of critical environmental conditions;
- Definition and control of critical limits for operation;
- Construction and qualification for adverse conditions.

For the full extensive descriptions of each OSO, the reader is referred to the original document. As an example, OSO#1 (Competent or approved operator) is described in more

detail: For low integrity, the operator has to be informed about each flight of the UAS. Checklists, maintenance and training procedures and a clearly formulated responsibility and duty distribution have to be established. The ConOps have to be specified. For a medium or high integrity, the operator has to be appropriate for the mission. This means that the complexity of the mission can be handled with the operator's resources. The operator has to be capable of identifying risks and performing means of mitigation. Before the first flight, an audit of a third party is required.

- Semantic data impact on risk assessment.

The actual flight trajectory cannot always be optimized only with respect to mitigating the impact of accidents or crashes. This is due to using—in most cases—only spatial data (e.g., like object geometry) for planning the flight trajectory. Implementing spatial data does not deliver the sufficient information needed for a safe path planning, navigation, and a reasonable risk assessment, for, e.g., a sufficient description of places is not always possible. For this reason, there are also other ideas of using semantic data to characterize objects and actions in the UAS environment related to operation. Semantics describe the object structure of an environment including contextual information, attributes, and their interrelationships, that is, non-spatial data. Exploiting semantic information beside spatial data in UAS operation can, on the one hand, improve the operational aspects and the time complexity of the planning process, which otherwise could be undesirably high. On the other hand, semantics can provide meaningful information for an enriched description of the current status of the scene and related constraints and rules that should be respected during an UAS operation.

To express the impact of semantic information on UAS path planning and risk assessment, we point to a practical example: the contextual term "bridge" indicates that this object is an important topological (traffic) connection between two regions. From a geometrical point of view, it might indicate that the bridge connects two shores of a river. In order to illustrate the dynamic character of semantic information, the following example is provided: The bridge is crossing the shortest path between two nodes on the trajectory of a UAV flight, but during rush hour it might be advisable to cross the river at a certain distance to the bridge in order to mitigate the risk for vehicles/people crossing the bridge, keeping in mind that additional energy must be spent. However, at certain times, outside of rush hour, it might be more suitable to cross the river closer to the bridge in order to save time and energy, knowing that the risk for objects on the ground is minimal. In short, the geometric and (dynamic) semantic representation of the UAS environment have to be coherently structured with a link in-between to ensure that consistent datasets form a convenient data source for mission planning to infer information from the urban environment of UAS. The benefit expected is to have the potential of providing a high level of safety in UAS management system through geometry and semantics-based definition of clear and simple rules that help operators and authorities in operating UAS as safely as possible. Though such kind of smart trajectory planning is—to the best of our knowledge—not available yet, it might have impact on the actual, individual risk assessment.

## 5. Conclusions

In summary, the new EU regulations provide detailed guidelines of how to define operations, identify risks and analyze situations prior to the deployment of UAS. An extensive documentation is required, depending on the risk and robustness of the planned mission, the UAS, and the operator. Safety measures include technical documentation and checks, as well as crew training and the analysis of the situation of the operation. For complex missions, the development towards standardized procedures and documentations such as those in manned air traffic helps to obtain flight permissions from the corresponding authorities. From a commercial point of view the regulations bring a certain level of reliability into economic considerations—it is worth for UAS producers to invest into safety measures in order to obtain a certain SORA classification.

**Author Contributions:** Conceptualization, A.A., A.L. and M.G.; validation, A.A., A.L.; formal analysis, A.A., A.L. and M.G.; data curation, A.A., A.L.; writing—original draft preparation A.A.; writing—review and editing, A.A., A.L. and M.G. All authors have read and agreed to the published version of the manuscript.

**Funding:** This research received no external funding.

**Institutional Review Board Statement:** Not Applicable.

**Informed Consent Statement:** Not applicable.

**Data Availability Statement:** Not Applicable.

**Acknowledgments:** The authors would like to thank Konrad Bärfuss, Lutz Bretschneider, Emily Hoffmann and Philipp Eberhardt for their contributions to the SORA classification of ALADINA and other UAS of the Institute of Flight Guidance, TU Braunschweig.

**Conflicts of Interest:** The authors declare no conflict of interest.

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
