# Peer review of "An Exploratory Investigation of UAS Regulations in Europe and the Impact on Effective Use and Economic Potential"

_drones, doi:10.3390/drones5030063_

Round 1
Reviewer 1 Report
This manuscript has been prepared well, the organisation and presentation of the content is excellent and informative. I have no suggestions to enhance the manuscript further, and look forward to seeing it published.
Author Response
Point1: the reviewer found the manuscript well prepared. He/she saw that the organisation and presentation of the content is excellent and informative.
Response1:
The authors thank the reviewer for taking the time to check the manuscript and write this review. According to your notice about the English language, we checked and edited the manuscript carefully.
Reviewer 2 Report
I think this is a fine commentary on the impact of the standardization of UAS operation regulations in the EU. However, this is more of a policy review than a research article.
The work does not present anything novel regarding the use and application of UAS, and does not fit within the aims and scope of the journal. Perhaps a different journal that focuses more on regulation and compliance issues of UAS and similar technologies would be more appropriate.
Author Response
Point1: I think this is a fine commentary on the impact of the standardization of UAS operation regulations in the EU.
Response1: We agree with the reviewer that the manuscript introduces a sound overview and fine commentary of UAS regulation in EU.
Point2: However, this is more of a policy review than a research article. The work does not present anything novel regarding the use and application of UAS, and does not fit within the aims and scope of the journal. Perhaps a different journal that focuses more on regulation and compliance issues of UAS and similar technologies would be more appropriate.
Response2:
We disagree that the work does not present anything novel, the question here how to interpret the word “novel”. In the manuscript we do not talk about developing of a software, tool, prototype, etc. The focus of this manuscript is on UAS regulations, which are new (adapted Jan 2021) and make a big difference in UAS sectors; and it is clear that these regulations lead to great changes in UAS operation. To the best of our knowledge, there are minor publications that discuss how these rules will affect UAS development/use and what about the challenges that could be arisen, because the regulations are new and still in a developing process. From this point of view, we came up to the idea of this paper in providing a comprehensive overview about the new regulations and linking to a practical example.
The reviewer found that the work does not fit to the aims and scope of the journal and advised to publish the manuscript in a different journal. To refute this, we refer to the journal aims and scope (journal homepage). One can see that the journal focuses on drone regulations and economic impact, and these aspects have been presented by our manuscript.
Point3: Notice about the English language
Response3: According to your notice about the English language, we checked and edited the manuscript carefully.
Reviewer 3 Report
The availability and operations of unmanned aircraft systems (UAS) in the national airspaces are growing fast. They bring along concerns regarding the safety of these applications, which called for new operational rules to mitigate the unavoidable risks of operations in civil airspace. The rapid technological development of UAS has been followed by new regulation rules that have constantly been changed from 2017 through 2020 to adapt to the evolution of new UAS technology. The present paper addresses the new European Union regulations for UAS operation starting on January 2021. The report presents a comprehensive overview of the new rules compared with existing regulatory laws before January 2021. These discussions are illustrated with a case study for the regular operation of the ALADINA UAS under the new EU operational rules. A clear understanding of these new rules are of great importance to the applications of UAS in the national civil airspace. The work presents the development of UAS regulations both on a global scale and within the EU. The different operational categories of UAS are presented, and detailed guidelines are provided regarding the UAS registration, operation restriction, and risk assessment, including technical documentation, pilot training, and safety procedures. The information presented in the paper provides a clear path to UAS operators to obtain a given SORA classification.
In line 80 (second page) please consider to substitute 'years. E.g.' for 'years, e.g.'.
Author Response
Point1: The reviewer found that the information presented in the paper presents a comprehensive overview of the new rules compared with existing regulatory laws before January 2021. It provides a clear path to UAS operators to obtain a given SORA classification.
Response1: The authors thank the reviewer for taking the time to check the paper and write this review. According to your notice about the English language, we checked and edited the manuscript carefully.
Round 2
Reviewer 2 Report
If the editors agree that this paper is in scope with the journal under regulations and economic impact, then I recommend for publication.
I do not have any issues with the quality or soundness of the manuscript.